# Analysis of Human Milk Microbiota in Northern Greece by Comparative 16S rRNA Sequencing vs. Local Dairy Animals

**DOI:** 10.3390/nu16142175

**Published:** 2024-07-09

**Authors:** Margaritis Tsifintaris, Michail Sitmalidis, Maria Tokamani, Christina Anastasiadi, Maria Georganta, Ilias Tsochantaridis, Dimitrios Vlachakis, Panagiotis Tsikouras, Nikolaos Nikolettos, George P. Chrousos, Raphael Sandaltzopoulos, Antonis Giannakakis

**Affiliations:** 1Department of Molecular Biology & Genetics, Democritus University of Thrace, 68100 Alexandroupolis, Greece; mtsifintaris@gmail.com (M.T.);; 2Department of Obstetrics and Gynecology, Democritus University of Thrace, 68100 Alexandroupolis, Greece; 3Department of Biotechnology, Agricultural University of Athens, 11855 Athens, Greece; 4University Research Institute of Maternal and Child Health and Precision Medicine, National and Kapodistrian University of Athens, 11527 Athens, Greece; 5UNESCO Chair of Adolescent Health, National and Kapodistrian University of Athens, 11527 Athens, Greece

**Keywords:** 16S rRNA amplicon sequencing, metagenomics, milk microbiota, human milk, bovine milk, goat milk

## Abstract

Milk is a biological fluid with a dynamic composition of micronutrients and bioactive molecules that serves as a vital nutrient source for infants. Milk composition is affected by multiple factors, including genetics, geographical location, environmental conditions, lactation phase, and maternal nutrition, and plays a key role in dictating its microbiome. This study addresses a less-explored aspect, comparing the microbial communities in human breast milk with those in mature milk from species that are used for milk consumption. Since mature animal milk is used as a supplement for both the infant (formula) and the child/adolescent, our main aim was to identify shared microbial communities in colostrum and mature human milk. Using 16S rRNA metagenomic sequencing, we focused on characterizing the milk microbiota in the Northern Greek population by identifying shared microbial communities across samples and comparing the relative abundance of prevalent genera. We analyzed ten human milk samples (from five mothers), with five collected three days postpartum (colostrum) and five collected thirty to forty days postpartum (mature milk) from corresponding mothers. To perform an interspecies comparison of human milk microbiota, we analyzed five goat and five bovine milk samples from a local dairy industry, collected fifty to seventy days after birth. Alpha diversity analysis indicated moderate diversity and stability in bovine milk, high richness in goat milk, and constrained diversity in breast milk. Beta diversity analysis revealed significant distinctions among mammalian species, emphasizing both presence/absence and abundance-based clustering. Despite noticeable differences, shared microbial components underscore fundamental aspects across all mammalian species, highlighting the presence of a core microbiota predominantly comprising the Proteobacteria, Firmicutes, and Actinobacteriota phyla. At the genus level, *Acinetobacter*, *Gemella*, and *Sphingobium* exhibit significant higher abundance in human milk compared to bovine and goat milk, while *Pseudomonas* and *Atopostipes* are more prevalent in animal milk. Our comparative analysis revealed differences and commonalities in the microbial communities of various mammalian milks and unraveled the existence of a common fundamental milk core microbiome. We thus revealed both species-specific and conserved microbial communities in human, bovine, and goat milk. The existence of a common core microbiome with conserved differences between colostrum and mature human milk underscores fundamental similarities in the microbiota of milk across mammalian species, which could offer valuable implications for optimizing the nutritional quality and safety of dairy products as well as supplements for infant health.

## 1. Introduction

The human milk microbiome represents a major source of microbial exposure for the breastfed infant and is an important driver of infant growth and development by beneficially impacting infant gene expression and immune development [1]. Although human breast milk is considered a hallmark of maternal-to-infant nutrition, milk from other animals, such as bovines and goats, also contributes significantly to human development. Across various species, milk is a complex and nutritionally rich fluid known for its essential role in nourishing infants and young children [2]. Beyond its well-recognized macronutrient content, it harbors a rich microbial community that plays a pivotal role in the development and well-being of infants and young children. The human microbiome comprises highly dynamic microbial communities inhabiting various body sites and engaging in intricate host–microbial interactions that display territory-specific complexity [3]. Recently, an increasing number of studies have reported the presence of many health-promoting bacteria (probiotics) in breast milk, including colostrum and mature milk, as well as bacteria typically considered pathogenic and frequently found in healthy controls [4]. Understanding the microbial similarities and dissimilarities across different mammalian milks as they compare to the human metagenome can provide valuable insights into the evolutionary constraints influencing milk’s microbiota stability and variability and their potential implications for human health. In addition, an interspecies comparison of human colostrum and mature milk, individually, could provide the conserved similarities that each maturation stage exhibits with the mature milk of the main dairy animals used for animal milk consumption.

To date, three distinct concepts have been proposed to elucidate the mechanisms through which these microorganisms reach human milk: the entero-mammary pathway, the retrograde inoculation pathway, and the notion of resident mammary microbiota [5]. The presence of microbial communities in colostrum even before the first infant feeding supports the entero-mammary pathway [6]. The vertical transmission of vaginal bacteria during childbirth may also influence milk microbiota [7]. Human breast milk bacteria contribute to the early colonization of the infant’s skin, nasal cavity, and gastrointestinal tract, conferring a head start to the host’s microbiota [8]. Bacteria from the child’s oral cavity, such as *Streptococcus*, *Rothia*, and *Gemella*, are commonly found in breast milk [9]. Additionally, commensal bacteria from the mother’s skin, including *Staphylococcus* and *Corynebacterium*, can colonize the mammary gland through the nipple [10]. However, it is important to note that the maternal skin microbiota and breast milk microbiota are distinct components of the overall microbiome [11]. Furthermore, anaerobic bacteria from the gastrointestinal tract, such as *Bifidobacterium*, *Bacteroides*, and *Clostridium*, are also present in breast milk [12,13]. Beyond colonization, breast milk harbors bacteria with probiotic properties, enhancing the neonate’s growth and well-being [14].

The composition of the human milk microbiome is influenced by various factors, including gestation length, area of residence, use of probiotics, intrapartum antibiotics, episodes of mastitis, and lactation stage. The gestation length is crucial, as a progressive increase in total bacterial load is associated with longer gestation periods [15]. Also, it has been shown that preterm milk contains lower levels of Bifidobacterium compared to term milk [16]. Furthermore, the stage of lactation affects the milk microbiota, with higher total bacterial loads observed in colostrum compared to mature milk and a greater relative abundance of anaerobic bacteria in mature milk [17]. Geographic location also significantly impacts the human milk microbiome, as variations in diet, hygiene practices, and environmental exposures contribute to differences in microbial communities [18,19]. The maternal consumption of probiotics during pregnancy has been linked to beneficial changes in the breast milk microbiome, increasing the detection rate of beneficial bacteria such as Lactobacillus, Bifidobacterium, and Streptococcus by 24% compared to that in controls [20]. Episodes of mastitis significantly alter the microbial composition of breast milk; during acute mastitis, Staphylococcus aureus dominates, whereas during subacute mastitis, Staphylococcus epidermidis is more prevalent [21]. The use of intrapartum antibiotics has been reported to lower the total bacterial load but increase alpha diversity and richness in the breast milk microbiota [22,23].

The development of the gastrointestinal tract in neonates is profoundly influenced by both intrinsic and extrinsic factors, among which diet plays a pivotal role. The human milk microbiome has been described as a dynamic ecosystem that is quite stable over time but also exhibits some variability between mothers based on their characteristics. A complex interplay between diet, infant host transcriptional responses, and gut microbiome composition may act as a key modulator of host–microbiome interactions, thereby influencing the trajectory of gut maturation in neonates [24]. Simultaneously, the interplay between colostrum and mature milk is intricate, marked by distinct differences, while at the same time, a “core” microbiome is temporally conserved, demonstrating considerable stability.

The analysis of breast milk bacteria was initially performed using traditional cultural methods, but advancements in DNA and RNA sequencing technologies have revolutionized research and diagnostics [10]. Of note, the rapidly evolving technological landscape evolved further with the advent of next-generation sequencing (NGS) technologies, which allow for rapid, high-throughput, and cost-effective sequencing and have led to major advances in our understanding of milk formation and effects. Metagenomics, a powerful approach within genomics, involves studying entire microbial communities directly from environmental or biological samples without the need for cultivation [25]. In breast milk research, 16S rRNA sequencing, a type of amplicon-based sequencing, is often employed to target the bacterial community composition, providing insights into taxonomic diversity. Over time, diverse methodologies have been employed for the characterization of the milk microbiome, producing comparable yet divergent outcomes owing to the inherent correlation between the maternal and infant microbiota.

The human milk microbiomes of bovines, goats, and humans have been extensively researched in numerous studies, highlighting the high abundance of Firmicutes, Actinobacteria, and Proteobacteria in the all three species [5,26,27]. These phyla commonly correspond to genera such as *Streptococcus*, *Pseudomonas*, and *Acinetobacter*. Nonetheless, more research needs to be conducted to directly compare the microbial compositions of the mature milk of dairy animals, which is commonly consumed by both infants and young children, within the same region and under the same study conditions. Our primary goal is to elucidate the core and conserved microbiomes of colostrum and mature human milk, using mature dairy animal milk as a biological filter. This study aims to characterize the human milk microbiota in relation to commonly consumed milk sources, particularly mature bovine and goat milk, which are frequently used as supplements in infant formulas and for child/adolescent nutrition. Using rigorous amplicon sequencing analysis, we seek to investigate the microbial compositions in each species, highlighting both the conserved similarities and divergences. These efforts are intended to contribute to a deeper understanding of the milk microbiota and its implications for human health.

## 2. Methods

### 2.1. Milk Sample Collection

Healthy lactating mothers were recruited at the University Hospital of Alexandroupolis (Greece). In total, ten samples were collected from five mothers (Table 1A), with each mother providing two samples at two different time points: colostrum (3 days postpartum) and mature milk (30–40 days postpartum). All participants delivered at full term (gestational age > 37 weeks) without pregnancy complications such as gestational diabetes (GD) or preeclampsia (PE), and no probiotic supplements were used during pregnancy or throughout the study period. All mothers were from the same region, and they also shared the same ethnicity. Simultaneously, women afflicted with bacterial infections, such as mastitis, as well as those undergoing antibiotic treatment, were excluded from this study. Before sample collection, the breast was cleaned, and the nipple was disinfected to avoid contamination from the infant’s mouth and surrounding skin, which could potentially skew the results and misrepresent the core milk microbiome. The collection was performed during the first morning feeding session, where the newborn was breastfed from one breast while a sterile breast pump was used to collect milk from the other breast. The breast milk samples were collected into sterile tubes. All participants provided written consent prior to enrollment, and the study was approved by the Ethics Committee of Democritus University of Thrace (protocol code 6111/30, date of approval 27 September 2021).

Bovine and caprine samples were also collected in the context of collaboration with the local dairy industry “Evrofarma” (Table 1B). The animals were sourced from the same farm and regularly checked by the company for health status, ensuring a consistent baseline. All samples were collected 50–70 days postpartum from Holstein cows and Saanen goats after their first lactation period. Each animal group (bovine and goat) followed the same feeding and breeding program. The udder was sterilized, and milk samples (ranging from 2–20 mL) were collected aseptically using a manual pump in 50 mL sterile tubes and kept at 4 °C during their transportation to the lab.

### 2.2. DNA Extraction, Library Preparation, and 16S RNA Ion Torrent Sequencing

A total number of 20 milk samples from 3 species were analyzed using Ion Torrent technology. A 1 mL aliquot of each milk sample was centrifuged at 16,000× *g* for 3 min at room temperature to pellet bacterial cells. DNA was extracted via mechanical lysis from the sample’s pellets using the Dneasy^®^ PowerSoil Kit (Qiagen, Tokyo, Japan) following the manufacturer’s instructions. DNA was quantified using the Qubit™ dsDNA HS Assay Kit (Cat: Q32854, Thermo Fisher Scientific, Waltham, MA, USA) with a Qubit 4.0™ Fluorometer (Cat. No. Q33226, Thermo Fisher Scientific, Waltham, MA, USA). The V4 variable region of the 16S rRNA gene, as a bacterial gene marker, was amplified with modified versions of the 515f [5′-GTGYCAGCMGCCGCGGTAA-3′] and 806 r [5′-GGACTACNVGGGTWTCTAAT-3′] primers using 5 ng of DNA and the KAPA SYBR Fast Master Mix (2×) Universal Kit (Cat no: KK4601, Sigma-Aldrich, St. Louis, MO, USA) in a total volume of 20 μL. The PCR reaction was performed using the following conditions: 95 °C for 3 min followed by 35 cyclesof amplification (denaturation: 95 °C for 15 s, annealing: 58 °C for 30 s, extension: 65 °C for 30 s). The PCR products were purified using the NucleoMag kit (Cat No.: 744970.50, Macherey-Nagel, Duren, Germany) and quantified using the Qubit™ dsDNA HS Assay Kit. One hundred nanograms of PCR product were used for library construction with the NEBNext^®^ Fast DNA Library Prep Set for Ion Torrent (Cat. No. E6270L, Thermo Fisher Scientific). According to the manufacturer’s instructions, the amplicons were tagged using the Ion Xpress Barcodes Adapters™ (Cat. No. 4474517, Thermo Fisher Scientific). Libraries were quantified using the Ion Library TaqMan^®^ Quantitation Kit (Cat No.: 4468802, Thermo Fisher Scientific, Waltham, MA, USA), diluted to a concentration of 100 pM, and then pooled. Emulsion PCR and template preparation for sequencing were performed on the Ion OneTouch 2 System with Ion 520™ & Ion 530™ ExT Kit (Cat No.: A27751, Thermo Fisher Scientific, Waltham, MA, USA) according to the manufacturer’s instructions. Sequencing was conducted using the Ion Torrent S5 system on a 530 chip (Cat No.: A27764, Thermo Fisher Scientific, Waltham, MA, USA).

### 2.3. Bioinformatics and Statistical Analysis

The initial steps involved using Torrent Suite by ThermoFisher Scientific with default settings for base calling and quality checking. All sequence data were exported in fastq format using samtools. The python script Split_on_Primer (https://github.com/Y-Lammers/Split_on_Primer (accessed on 2 December 2023)) was utilized to sort reads based on the PCR primers used, with a permissible tolerance of up to two mismatches. Further analysis was conducted using Qiime2-2021.8 [28]. Cutadapt [29] was used to exclude any reads shorter than 150 bp. DADA2 was used to denoise the reads using the pseudo-pooling method. Quality-score filtering, truncation, and trimming were also performed using DADA2. Reads were truncated to their expected length based on their quality score and were later trimmed 15 base pairs from the left to remove PCR primers. The de novo consensus method was conducted to remove any chimeric sequences, with an error model that had been trained on a subsample of 1,000,000 reads. The taxonomic assignment of ASVs was performed by a pre-trained classifier (Silva.v138 [30]) in conjunction with the BLAST+ algorithm with default parameters and a coverage criterion of ≥0.95. Relative abundance bar plots as well as UpSet plots were generated using the ggplot2 and upSetR R packages. Each sample’s alpha diversity was calculated to investigate the composition of milk produced by different species, using five different indices from the ASV table produced earlier. These included Simpson, Chao1, Pielou’s evenness, and Shannon’s indices. The statistical significance of differences between the species was calculated using the Kruskal–Wallis test. Alpha diversity boxplots were generated using the ggplot2 package in R. To investigate the differences between the sample communities, four different indices, weighted and unweighted UniFrac, Jaccard, and Bray–Curtis, were calculated to study beta diversity. Principal coordinates analysis (PCoA) and non-metric multidimensional scaling (NMDS) analysis were performed to better visualize the differences between the bacterial communities of our samples, and PERMANOVA was used to assess their statistical significance. The filtering step included omitting any bacteria that was not detected in four or five samples from each group (bovine, goat, human colostrum, and mature human milk). This step guaranteed that any observed overlap between groups would not be compromised by the inclusion of bacteria exhibiting irregular occurrence patterns across samples. In addition, samples that did not meet the sequencing depth criteria to be identified at the genus level were excluded from the analysis, along with any uncultured bacteria that have not been identified. Differential abundance analysis (DAA) was conducted to identify differences in the abundances of bacterial genera between animal and human milk; human colostrum (human-A) and mature milk (human-B) were analyzed against goat and bovine mature milk. The DESeq2 [31] package, which consists of an over-dispersed count model and performs relative log expression (RLE) normalization, was used in this study. Also, we utilized the MAMI [32] and Peryton [33] databases to extract information concerning the bacteria that were detected in the milk samples and their associations with various diseases.

## 3. Results

### 3.1. Sample Characteristics and 16S rRNA Sequencing Data Description

This study focused on homogeneity across several key characteristics to conduct a robust comparative analysis of milk microbiomes. A detailed summary of all milk samples, including demographic and anthropometric data, is presented in Appendix A. Metagenomic sequencing returned a total of 1,237,761 reads distributed among 20 samples and four groups: bovine (N = 5), goat (N = 5), human-A (colostrum milk, N = 5), and human-B (mature milk, N = 5). As depicted in Appendix A, the highest number of reads was noticed in the human-A group (400,493 reads) and the lowest in the human-B group (209,345). In contrast, the bovine and goat groups exhibited similar numbers of reads (304,364 and 323,559, respectively). Most of these reads were assigned to amplicon sequence variants (ASVs) at the following taxonomic ranks: phylum, class, order, family, genus, and species. Up to the genus level, 96–99% of the reads were assigned to ASVs, while at the species level, an average of 52.3% of reads were assigned (Appendix A).

### 3.2. Diversity of Bacterial Communities

To estimate the mean bacterial diversity of milk’s microbial communities within each species, the standard indices for evenness and richness were calculated (Figure 1 and Appendix A). Alpha diversity, as measured by the Chao1 index, showed significant differences between all groups (bovine vs. goat; *p* = 0.028, goat vs. human; *p* = 0.09, bovine vs. human; *p* = 0.09, Kruskal–Wallis test), with goat milk having the highest value, indicating more bacterial communities (Figure 2A). Simpson’s diversity index (Figure 1B), which accounts for both species number and relative abundance, indicated higher diversity in goat and human-B samples, suggesting the dominance of certain bacterial genera. Significant differences were observed between bovine and goat milk (*p* = 0.009), bovine and human mature milk (*p* = 0.009), and goat and human milk (*p* = 0.047). The higher Shannon index in goat milk underscores the presence of more stable microbial communities with a more even distribution (Figure 2C). Significant differences were observed between the goat group and the other two groups (human and bovine, *p* = 0.09). Human and bovine milk showed no statistical difference (*p* = 0.602) in this diversity index. Evaluating diversity alongside species richness, Pielou’s evenness showed higher but distinct indices in goat and human mature milk (*p* = 0.028), implying relatively even bacterial compositions (Figure 1D). In contrast, bovine milk had a lower score, suggesting the dominance of certain bacterial species within the samples (Bovine vs. Goat; *p* = 0.009, Bovine vs. Human; *p* = 0.016). Overall, the results indicate that goat milk has the highest alpha diversity compared to bovine and human mature milk.

A similar approach was followed to compare human colostrum (human-A) and mature milk (human-B). Colostrum milk exhibited higher values across Shannon entropy, Chao1, and Simpson index, indicating greater bacterial diversity and abundance (Appendix A). However, no significant difference was observed in Pielou’s evenness, suggesting that the evenness of bacterial distribution remains relatively stable between colostrum and mature milk.

Principal coordinates analysis (PCoA) was used to thoroughly investigate the beta diversity in milk sample groups, illustrating variations in microbial community composition (Figure 2). The results revealed distinct clustering patterns for bacterial communities among the three species examined. However, a notable finding was the high variability observed between colostrum and mature milk within human samples, indicating a dynamic microbial transition during lactation.

### 3.3. Microbial Community Composition in Milk Samples

A taxonomic assignment of amplicon sequence variants (ASVs) detected in milk samples from bovine, goat, and human samples (human-A: colostrum milk, human-B: mature milk) is presented in Figure 3. At the phylum level, as depicted in Figure 3A, goat milk exhibited the highest diversity, with a total number of 24 ASVs, followed by bovine milk with 19 ASVs, human colostrum milk with 13 ASVs, and human mature milk with 11 ASVs. At the order level, goat milk showed again the greatest diversity with 93 ASVs, while bovine milk exhibited 78 ASVs, human colostrum milk 51 ASVs, and human mature milk 39 ASVs. In total, 302 genera were detected in goat milk, 207 in bovine milk, 124 in human colostrum, and 88 in human mature milk.

Taxonomic assignment results showed the major presence of five phyla, 15 orders, and 26 genera across all milk samples (>1% relative abundance), as illustrated in the stacked bar graph in Figure 3B. Proteobacteria and Firmicutes were the most common phyla across all studied species. Proteobacteria were detected with relative abundances of 52.01% in bovine milk, 19.13% in goat milk, 45.19% in human colostrum milk, and 41.88% in human mature milk. Actinobacteriota and Bacteroidota showed a notable presence, especially in goat milk, with relative abundances of 18.98% and 13.38, respectively. Also, Euryarchaeota were found in lower proportions in human milk (<1%) compared to animal species (bovine and goat). At the order level, Pseudomonadales and Lactobacillales were predominant in bovine milk, with relative abundances of 49.73% and 34.02% respectively. In goat milk, Staphylococcales (22.72%) and Pseudomonadales (20.06%) were found to be the most abundant. Human colostrum milk was characterized by high proportions of Staphylococcales (23.82%) and Lactobacillales (20.06%), whereas human mature milk was dominated by Staphylococcales (26.79%), Pseudomonadales (24.99%), and Lactobacillales (22.1%). Significant genera detected across all samples included *Actinobacter*, *Streptococcus*, *Staphylococcus*, *Gemella*, *Jeotgalicoccus*, *Pseudomonas*, and *Corynebacterium* (Appendix A). In bovine milk, *Acinetobacter* (41.18%) and *Streptococcus* (30.41%) were notably abundant, while goat milk showed higher levels of *Staphylococcus*, with a relative abundance of 20.23%. Human colostrum milk was rich in *Streptococcus* (19.48%), *Sphingobium* (16.84%), *Actinobacter* (15.15%), *Gemella* (13.07%), and *Staphylococcus* (10.71%). These genera were also detected in human mature milk; however, *Actinobacter* was the most abundant (24.58%), followed by *Streptococcus* (20.05%), *Gemella* (12.31%), *Staphylococcus* (12.45%), and *Sphingobium* (9.41%).

### 3.4. Core Milk Microbiome of Bovine, Goat, and Human Milk

The microbial compositions of bovine, goat, and human milk (A: colostrum and B: mature) were analyzed to identify core bacterial genera in the colostrum and mature human milk. For inclusion in this analysis, genera had to be detected in at least four out of five samples within each group. The Venn diagrams in Appendix A (see also Appendix A) reveal the shared and unique bacteria among the milk samples across the four groups (bovine, goat, human colostrum, and human mature milk). At the phylum level (Appendix A), the comparative analysis of microbial diversity reveals that five phyla (*Proteobacteria*, *Firmicutes*, *Actinobacteriota*, *Bacteroidota*, and *Euryarchaeota*) constitute a core set of microbial phyla present in all milk types. These phyla are consistently found across bovine, goat, and human colostrum milk. However, *Euryarchaeota* is absent in human mature milk, indicating some variation in microbial communities between different stages of lactation. Additionally, three phyla (*Spirochaetota*, *Fusobacteriota*, *Chloroflexi*) were consistently detected in both bovine and goat milk but were not consistently detected in human milk. As shown in Appendix A, at the order level, a greater number of unique bacteria was detected in goat milk (19) compared to bovine (2) and human milk (1). Notably, 14 orders were common among all groups, while an additional 5 orders were detected consistently in bovine, goat, and human colostrum milk. Moreover, bovine and goat milk showed a significant overlap, with 10 bacteria (order level) detected in animal milk but not in human milk.

The distribution of bacterial genera is illustrated in the Venn plot in Figure 4A. A total of 133 genera were detected consistently, with a distinct and overlapping presence among the groups. In bovine milk, 10 distinct genera were detected, while goat milk exhibited 54 unique genera. Moreover, both human colostrum and human mature milk exhibited two distinct genera each. A core of 16 genera was shared among all four groups, indicating a common microbiota component in milk from different sources. Additionally, 15 genera were shared between bovine, goat, and human colostrum milk. The Sankey plot (Figure 4B) presents the core microbiota of human milk, highlighting the genera distribution between colostrum (human-A) and mature milk (human-B). Human colostrum milk contained 21 unique genera, while human mature milk contained 4 unique genera. Notably, 17 genera were detected consistently in both colostrum and mature milk, signifying a substantial overlap in the microbial communities at different lactation stages.

The radial network graph (Figure 4C) provides a detailed view of the associations between bacterial genera and the four groups. Genera are positioned around the circle, with connections to the relevant host groups indicated by color coding. Human colostrum milk (pink nodes) featured unique genera such as *Cellvibrio* and *Granukucatekka*, while human mature milk (purple nodes) included unique genera like *Blastomonas* and *Thermus*. The common genera observed across all groups (represented by red nodes), including *Acinetobacter*, *Corynebacterium*, *Gemella*, *Staphylococcus*, *Streptococcus*, and *Pseudomonas* as some of the most prevalent bacteria, represent the core microbiota of milk. In summary, the comprehensive analysis revealed distinct and overlapping microbial communities in bovine, goat, and human milk, with a notable core microbiota shared among all groups.

### 3.5. Differential Abundance Analysis

To undertake a more detailed examination of the differences in the abundance of each taxon (genus level) among all groups, we performed a differential abundance analysis (results are presented in Appendix A). Figure 5 illustrates the analysis of core human milk bacterial abundance in comparison to that of local dairy animals. The genera *Micrococcus*, *Enterococcus*, and *Massilia* were detected consistently in human mature milk, bovine milk, and goat milk but not in human colostrum milk. Moreover, *Enterococcus* and *Micrococcus* were found at a statistically higher abundance than in bovine milk (FC = 4.16 and FC = 4.79), while *Enterococcus* also exhibited a higher abundance than in goat milk (FC = 7.36). In the context of human bacteria detected consistently only in colostrum milk, *Finegoldia* stands out as the sole genus exhibiting higher abundance. Specifically, *Finegoldia* demonstrated a fold change of 8.21 and 7.62 relative to bovine and goat milk, respectively. As for *Methanobrevibacter*, *Bacteroides*, *Romboutsia*, *UCG-005*, and *Halomonas*, they all exhibited lower abundances in human than in animal milk, with fold changes ranging from −4.07 to −5.56.

Bacteria that were detected in both colostrum and mature human milk presented distinct abundance profiles. *Pseudomonas*, *Psychrobacter*, *Jeotgalicoccus*, *Turicibacter*, and *Atopostipes* all displayed lower abundances in human milk compared to animal milk, with fold changes ranging from −2.43 to −6.88. Interestingly, *Carnobacterium* was detected at a higher proportion in bovine milk than human milk at both time points: colostrum milk (FC = −8.39) and mature milk (FC = −6.45), respectively. Conversely, some of the most abundant bacteria found in this study were detected more frequently in human milk. *Acinetobacter* and *Streptococcus* were found in greater proportions in human colostrum and mature milk than in goat milk, while *Staphylococcus* followed the same pattern, but in comparison to bovine milk. *Anaerococcus*, *Sphingobium*, and *Gemella* showed a statistically significant increase in human milk relative to levels in either of the dairy animals’ milk.

The same 39 bacteria were also investigated in the Microbiome Atlas of Mothers and Infants (MAMI) database so as to cross validate the occurrence of bacteria in breast milk, as well as identify any bacteria that have not been detected in breast milk before (Appendix A). Out of 39 bacterial genera, 32 genera (76.9%) were listed as existing in breast milk, while the other 7 had not been identified before (23.07%). More specifically, three genera (*Anaerococcus*, *Finegoldia*, *Romboutsia*) had been detected in other mother–infant tissues, while for four genera (*Atopostipes*, *Jeotgalicoccus*, *Ulvibacter*), there were no entries available in the database. To further our understanding, we searched the Peryton database to uncover diseases linked with alterations in microbial community abundances (Appendix A). Notably, a substantial number of identified illnesses and gastrointestinal complications like ulcerative colitis, Crohn’s disease, irritable bowel syndrome, and diarrhea showed associations with alterations in microbial abundance.

## 4. Discussion

Milk, while renowned for its proteins and lipids, also constitutes a dynamic blend of macronutrients, micronutrients, and bioactive molecules, not only serving as a reservoir of these essential components but also harboring a diverse microbial consortium with notable impacts on infant and child development [7,34,35]. The detection of over 1300 distinct species of bacteria in human milk highlights its rich microbial diversity. However, accurately defining the core microbiome of human milk remains contentious due to factors such as individual differences, geographical variations in studies, and variations in methods employed for milk collection, storage, and analysis [36]. Currently, there is no consensus on defining a widely accepted core milk microbiome that is considered healthy for infants and children [5].

The intricate interplay between breast milk microbiota and maternal tissues, including the skin, gastrointestinal tract, and the infant’s oral cavity, underscores the inherent variability in raw milk samples [37]. Even among milks sourced from animals of identical breeds subjected to the same diet, notable distinctions emerge, reflecting the multifaceted factors shaping milk microbiota. The variation in dietary patterns and lifestyles among human populations further complicates efforts to pinpoint exact determinants influencing milk microbiota, as samples can differ vastly [38]. This observation sheds light on the disparities in findings among studies in the field of milk microbiota, highlighting the complexity that still characterizes this area of research. The present study endeavors to shed light on the unique differences among human breast milk, cow bovine milk, and goat milk while also identifying commonalities to establish the fundamental microbiome of colostrum and mature human milk that is conserved in the main kinds of mature animal milk used for consumption.

Examining alpha diversity, low values across all four indices in bovine milk indicate a state of moderate diversity with specific genera’s prominence. This suggests established and stable microbial communities, highlighting niche-specific adaptation. In contrast, goat milk displays consistently high values across all four indices, emphasizing richness and even distribution. This substantiates the heightened adaptation of the goat milk microbiota to its specific ecological niche, fostering purposeful interactions among bacteria. The findings strongly indicate a state of high stability and adaptability within goat milk microbial communities, attesting to their resilience. Regarding beta diversity, distinctions among mammalian species are noted, with minimal dissimilarities between colostrum and mature human breast milk. This suggests potential common taxonomic lineages among bacterial genera, despite significant differences.

Exploring the relative abundance of milk microbiota at the phylum level, we noticed that *Proteobacteria*, *Firmicutes*, and Actinobacteria exist in comparable relative abundances in all groups. Many studies, including ours, have shown that *Firmicutes* and *Proteobacteria* were the predominant phyla in breast milk, with Actinobacteria and Bacteroidetes present at lower relative abundances [39,40,41,42,43]. These commonalities underscore fundamental microbial components in milk across species, emphasizing the potential importance of *Firmicutes* and *Proteobacteria* in shaping the shared characteristics of the mammalian milk microbiome. Several studies [44,45,46] support that an increase in the abundance of certain phyla in human breast milk are associated with a decrease in microbial diversity and species dominance. However, the transition from colostrum to mature milk reveals differences in *Actinobacteriota (Corynebacterium*) and *Firmicutes* (*Enterococcus*). These findings underscore intricate microbial dynamics in diverse milk types and across lactation stages [39].

Establishing the core composition of the milk microbiome is challenging due to the numerous factors that influence its modulation [47]. Variability within the human microbiota has been observed during the transition from colostrum to mature human milk [5]. Many bacteria detected in human colostrum but not in mature milk were found in higher abundances in bovine and goat milk, highlighting the differences in microbial compositions across species and lactation stages. This highlights the complex microbial environment in colostrum compared to mature milk. For instance, *Bacteroides* assists the host by fermenting dietary polysaccharides, while *Methanobrevibacter* consumes end-stage fermentation products, potentially relieving the feedback inhibition of upstream microbes. Together, their synergistic metabolic activity plays a vital role in human gut health, underscoring the importance of these microbial communities in early development [48]. This synergistic metabolic activity plays a vital role in human gut health, underscoring the importance of these microbial communities in early development. In contrast, bacteria of the genus *Finegoldia* are consistently detected at a higher abundance in human colostrum milk, indicating their potential role in early infant gut colonization. Furthermore, colostrum exhibits a higher concentration of nutrients and bioactive agents that may be utilized by microorganisms as sustenance [49,50]. The presence of this rich diversity of microbes, nutrients, and bioactive agents in colostrum highlights its crucial role in facilitating a newborn’s successful transition to life outside the womb in those early days. Bacteria such as *Micrococcus*, *Enterococcus*, and *Massilia*, detected in mature human milk but not in colostrum, are also consistently found in animal milk, suggesting a shared microbial environment. *Enterococcus*, a part of the gut commensal microbiota, was significantly more abundant in human milk compared to animal milk and is deeply involved in the complex web of metabolic interactions with other gut inhabitants and the host [51].

A substantial overlap in bacterial communities is observed in human colostrum, mature milk, and animal milk. Genera such as *Staphylococcus* and *Streptococcus*, which are among the ten most commonly reported bacterial species, have been consistently detected throughout all lactation phases [4,52,53]. These genera, primarily consisting of commensal bacteria found on human skin and mucosa, play crucial roles in early neonatal microbial colonization and immune system development. Despite their potential pathogenicity, these species play roles in early neonatal microbial colonization and immune system development. It has been shown that bacteria such as *Corynebacterium* and *Staphylococcus* can directly inhibit pathogen growth by producing antimicrobial substances, as well as by competing for nutrients and adhesion sites [54]. *Streptococcus* species, with their characteristic chain-like arrangement, contribute to regulating oral cavity pH and form biofilms that antagonize pathogenic bacteria [55]. Additionally, *Corynebacterium*, *Staphylococcus*, *Streptococcus*, *Gemella*, and *Shibgobium* are part of the core microbiome but were found in significantly higher abundance in human milk compared to animal milk. That could be due to their beneficial capabilities, such as inhibiting the growth of potentially pathogenic bacteria and regulating the skin microbiota, or their probiotic effects, which are considered essential for infant growth [56]. Most of these bacteria are commonly found in the gut of humans and bovines, which further confirms the existence of an entero-mammary pathway for maternal bacteria in both mammals, which has already been shown [13,57,58]. This viewpoint emphasizes the complex link between human milk composition and infant well-being, stressing the need to comprehend the evolutionary forces influencing the human milk microbiota for enhancing the health of mothers and infants [47].

During our investigation, we identified several bacterial genera that had not previously been detected in human breast milk. *Anaerococcus*, *Carnobacterium*, *Finegoldia*, and *Rombutsia* have been identified in mother–infant samples, but not in breast milk. Additionally, *Anaerococcus* and *Finegoldia* were mostly observed in vaginal samples, followed by gut and skin samples. At the same time, *Carnobacterium* has only been found in the gut, and *Romboutsia* is specific to the gut and vagina [32]. The presence of these microbial communities in various human tissues supports the concept that these tissues are interconnected while underscoring the significance of these bacteria for infant growth. On the other hand, no records were found for *Atopostipes*, *Chitinophaga*, *Jeotgalicoccus*, *Ruminococcaceae UCG-005*, and *Ulvibacter* in the Microbiome Atlas of Mothers and Infants database [32].

Regarding the association of bacteria found in human milk with diseases, the microbial communities of breast milk are often the same as those found in the infant gut, contributing to its development. Moreover, alterations in the gut microbiota can have a profound impact on the composition of human milk, consequently influencing the establishment of the infant microbiome [5]. An uneven distribution or dysbiosis in these microbial populations can lead to various health issues in infants. When digested by the neonate, certain microorganisms found in breast milk produce short-chain fatty acids, which are essential for regulating the immune system, lowering inflammation, protecting the colon, and acidifying the intestinal tract to prevent harmful bacteria growth. Thus, any imbalance in infant bacterial communities can potentially cause gut diseases or complications, which places further emphasis on breastfeeding and the importance of breast milk.

Although the limitations of 16S rRNA sequencing must be acknowledged, our findings provide valuable insights into the potential functional roles of specific bacteria within breast milk through a direct comparison with the bacteria found in the mature milk of local dairy animals. In our study, milk samples were collected from cleansed nipples (aseptic sampling) to minimize the risk of contamination from external, non-resident bacteria, including those from the infant’s mouth and surrounding skin. While this approach allowed us to obtain a clear and focused understanding of the microbial communities directly associated with breast milk, it should be noted that in a natural breastfeeding scenario, disinfection is not practiced, and the maternal skin microbiota also play a beneficial role in shaping the infant’s microbiota. Future studies should compare samples obtained under aseptic conditions with those collected without disinfection to elucidate the impact of everyday breastfeeding practices on the complex interactions and functional significance of microbial communities in shaping infant health and development.

## 5. Conclusions

This study investigates the milk microbiomes of human, bovine, and goat milk, revealing significant differences and commonalities. Goat milk exhibited the highest microbial diversity, with the most ASVs. Human colostrum demonstrated greater bacterial diversity and abundance compared to mature human milk. A core of 16 genera was shared among all groups, indicating a common microbiota component in the main milk sources for infants and children (breast milk colostrum/mature, bovine, and goat mature). Notably, the genera *Acinetobacter*, *Corynebacterium*, *Gemella*, *Staphylococcus*, *Streptococcus*, and *Pseudomonas* were prevalent across all groups, representing the core microbiota of milk. These findings provide a comprehensive analysis of the variability in the abundance of milk’s core microbiome in breast milk (colostrum and mature) compared to the main dairy animals’ mature milk. In addition, the successful direct comparison of human and animal milk from the same region may lay the groundwork for future larger studies at optimizing infants’ and children’s nutrition and health.

## Figures and Tables

**Figure 1 nutrients-16-02175-f001:**
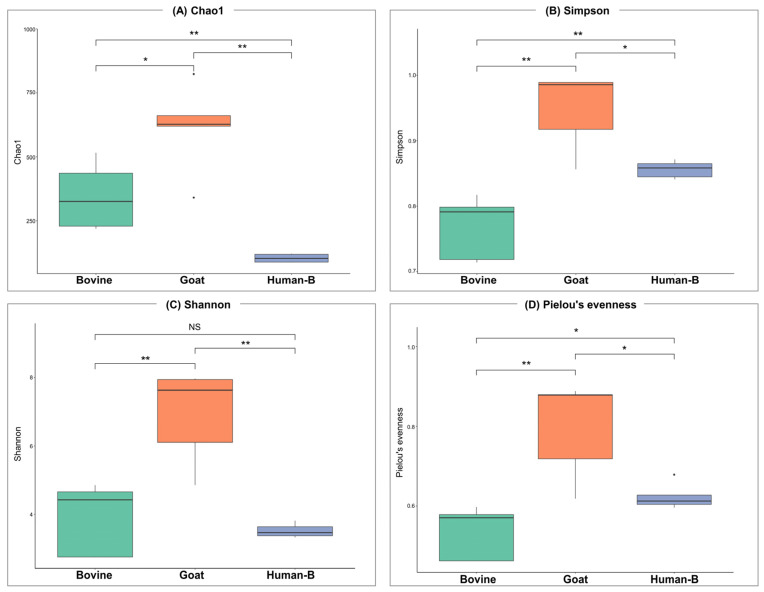
Bovine, goat, and human mature milk microbiome alpha diversity. (**A**) Chao1, (**B**) Simpson, and (**C**) Shannon indices and (**D**) Pielou’s evenness. Statistical significance (Wilcoxon test); NS *p* > 0.05, (*) *p* < 0.05, (**) *p* < 0.001.

**Figure 2 nutrients-16-02175-f002:**
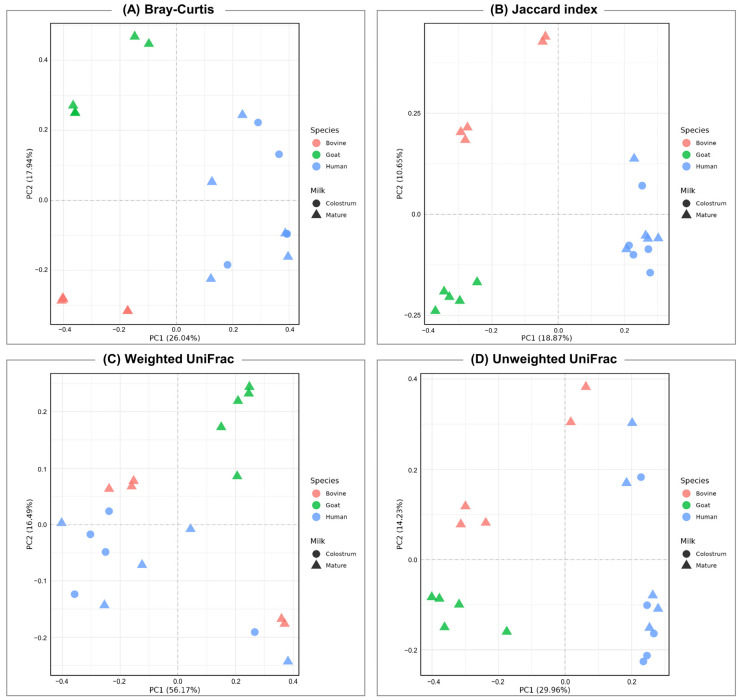
Principal coordinates analysis (PCoA) plot based on the (**A**) Bray-Curtis dissimilarity matrix (**B**) Jaccard index, (**C**) unweighted UniFrac distance, and (**D**) weighted UniFrac distance matrix, calculated for genus-level abundances and depicting patterns of beta diversity for microbial communities in milk.

**Figure 3 nutrients-16-02175-f003:**
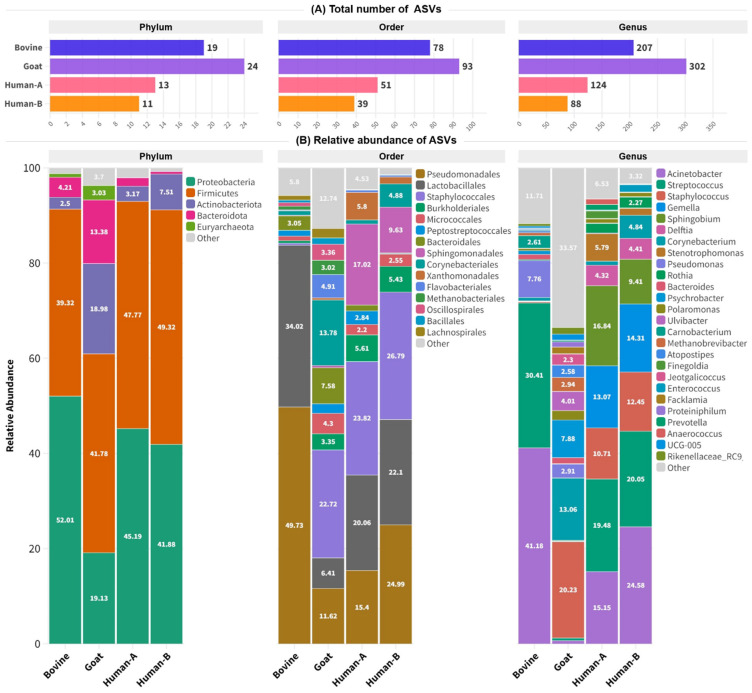
Taxonomic summary of amplicon sequence variants detected in milk samples. (**A**) Total number of ASVs at the phylum, order, and genus levels. (**B**) A stacked bar graph shows the relative abundance of microorganisms detected in milk samples. Relative frequencies smaller than 1% were collapsed into the “other” category. Human-A: colostrum milk; Human-B: mature milk.

**Figure 4 nutrients-16-02175-f004:**
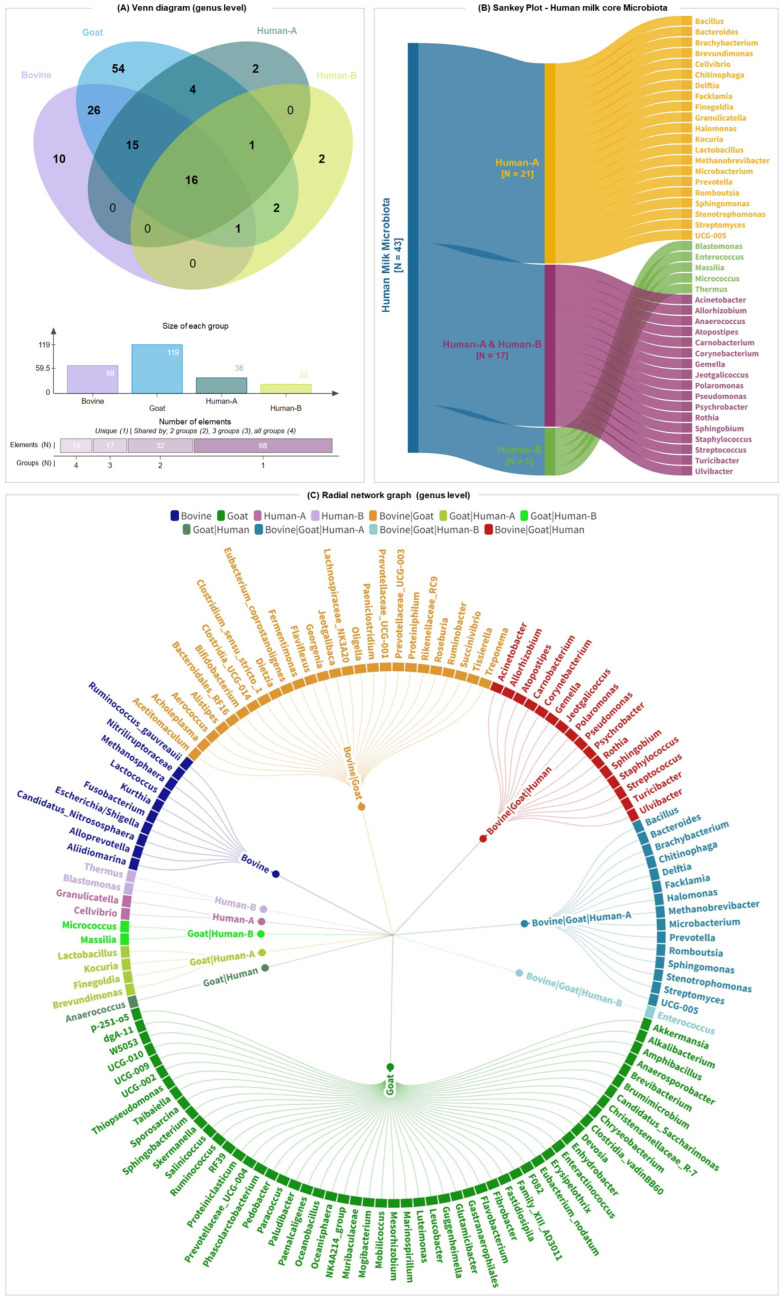
Comparison of core microbiota in bovine, goat, and human milk (colostrum and mature) at the genus level. (**A**) Venn diagram illustrating the numbers of shared and unique amplicon sequence variants between three species. (**B**) Sankey plot depicting the genera that are detected consistently in human breast milk samples in timepoint A (human-A: colostrum milk) and timepoint B (human-B: mature milk). (**C**) Radial network graph depicting genus-level overlap of bacterial communities in the milk microbiota of bovine, goat, and human milk. The detection rate threshold was set to 80% (4/5 samples).

**Figure 5 nutrients-16-02175-f005:**
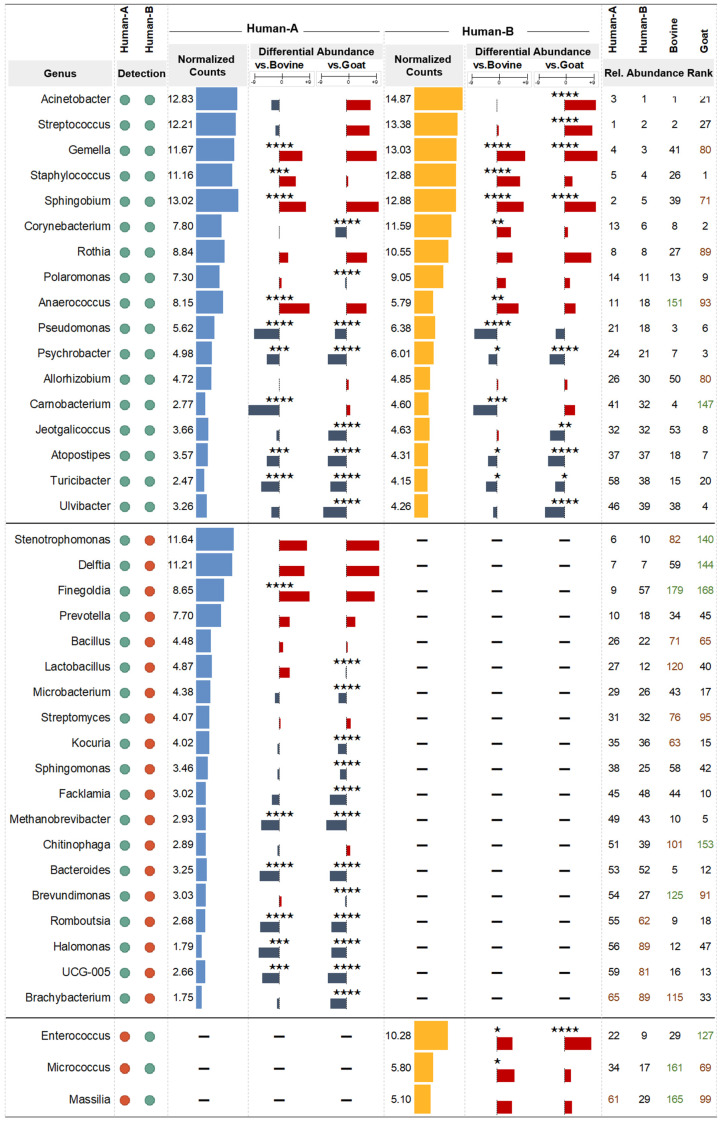
Comparative analysis of core milk bacterial abundance between humans and local dairy animals. Bacterial genera that met the detection rate threshold (80%; detected in at least four out of five samples) are depicted with green dots, whereas those that did not meet the threshold are indicated with red dots. Differential abundance results for genera that did not meet the detection threshold were not considered and are marked with a dash (“-”). Results with an adjusted *p*-value less than 0.05 were considered statistically significant (four stars: *p* < 0.0001, three stars: *p* < 0.001, two stars: *p* < 0.01, one star: *p* < 0.05, no stars: *p* ≥ 0.05). Relative abundance rank is indicated by color: high (1–60) in black, medium (61–120) in red, and low (>120) in green.

**Table 1 nutrients-16-02175-t001:** Milk samples analyzed with 16S metagenomic sequencing. A. Human breast milk samples. B. Animal milk samples.

	Individual ID	Sample ID	Day of Collection	Stage of Lactation	Species	Living Area
(A) Human Breast Milk Samples						
	H1	H1-A	3	Colostrum	Human	Thrace
	H1	H1-B	30–40	Mature	Human	Thrace
	H2	H2-A	3	Colostrum	Human	Thrace
	H2	H2-B	30–40	Mature	Human	Thrace
	H3	H3-A	3	Colostrum	Human	Thrace
	H3	H3-B	30–40	Mature	Human	Thrace
	H4	H4-A	3	Colostrum	Human	Thrace
	H4	H4-B	30–40	Mature	Human	Thrace
	H5	H5-A	3	Colostrum	Human	Thrace
	H5	H5-B	30–40	Mature	Human	Thrace
(B) Animal Milk Samples						
	C1	C1	50–70	Mature	Bovine	Dairy Industry
	C2	C2	50–70	Mature	Bovine	Dairy Industry
	C3	C3	50–70	Mature	Bovine	Dairy Industry
	C4	C4	50–70	Mature	Bovine	Dairy Industry
	C5	C5	50–70	Mature	Bovine	Dairy Industry
	G1	G1	50–70	Mature	Goat	Dairy Industry
	G2	G2	50–70	Mature	Goat	Dairy Industry
	G3	G3	50–70	Mature	Goat	Dairy Industry
	G4	G4	50–70	Mature	Goat	Dairy Industry
	G5	G5	50–70	Mature	Goat	Dairy Industry

## Data Availability

The data presented in this study have been deposited and are freely available in the European Nucleotide Archive (ENA) at EMBL-EBI under accession number PRJEB75815 (https://www.ebi.ac.uk/ena/browser/view/PRJEB75815, accessed on 3 July 2024).

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
