# Peer review of "Analysis of Human Milk Microbiota in Northern Greece by Comparative 16S rRNA Sequencing vs. Local Dairy Animals"

_nutrients, 2024, doi:10.3390/nu16142175_

Round 1

Reviewer 1 Report

Comments and Suggestions for Authors

The idea of â€‹â€‹the scientific study is very interesting, contributing a lot to science, but I have a few comments:

1. The abstract requires editing. The background should be shortened, while in terms of methodology it should be expanded; among others by providing the number of the study group, time of sample collection.

2. line 60: In the introduction, when describing the hypothetical sources of the human milk microbiome, please add bacteria found in human milk from the mother's skin, the child's oral cavity and the mother's digestive tract.

3. line 70: Is diet crucial for the human milk microbiome? Please provide references that confirm this statement.

4. The introduction does not mention factors differentiating the human milk microbiome, i.e.: length of pregnancy, area of â€‹â€‹residence, use of probiotics.

5. Why was the analysis of transitional milk omitted?

6. Lack of precise description of how the milk was obtained. Was the milk collected manually or with a breast pump?

7. Why was milk not collected from animals at the same time points as from women?

8. The issue of hygiene during the collection of material for testing is debatable. The level of hygiene in women is higher than in animals. Doesn't this affect the results of the study?

9. The study group is very small, I suggest conducting research on a larger scale. Was the possibility of the influence of factors such as: the woman's intake of probiotics, the length of hospitalization, diseases during pregnancy such as gestational diabetes, taken into account?

10. In the discussion, please do not repeat the description of your results, which have already been described earlier - in the results section.

Author Response

Comments 1. The abstract requires editing. The background should be shortened, while in terms of methodology it should be expanded; among others by providing the number of the study group, time of sample collection.

Response 1: We appreciate your feedback, and the recommended changes will provide a clearer and more comprehensive overview of our study. In response to your suggestion, we have revised the abstract by shortening the background section and expanding the methodology section. We have included more detailed information regarding the study group size and the sample collection (revised lines; 17-28 and 36-42)

Comments 2. line 60: In the introduction, when describing the hypothetical sources of the human milk microbiome, please add bacteria found in human milk from the mother's skin, the child's oral cavity and the mother's digestive tract.

Response 2: Thank you for the recommendation. We have included the information (lines: 82-102) about bacteria found in human milk from the mother's skin, the child's oral cavity, and the mother's digestive tract, along with the corresponding references. In child's oral cavity; Streptococcus, Rothia, and Gemella (DOI: 10.3389/fmicb.2018.02512), in mother's skin; Staphylococcus and Corynebacterium (10.1093/nutrit/nuaa029) and in digestive tract; Bifidobacterium, Bacteroides, and Clostridium, (DOI: 10.1111/1462-2920.12238, DOI: 10.3945/an.114.007229).

Comments 3. line 70: Is diet crucial for the human milk microbiome? Please provide references that confirm this statement.

Response 3: Thank you for pointing out the need for references regarding the influence of diet on the human milk microbiome. Although few studies have investigated this topic, existing research provides evidence of diet's impact on the composition of breast milk microbiota.

For instance, calorie intake has been positively associated with the relative abundance of Granulicatella (DOI: 10.1002/mnfr.201700600). Moreover, Human Milk Oligosaccharides (HMO) serves as a metabolic substrate for desired bacteria and were found to be dependent on factors such as body mass index (BMI) (DOI: 10.1093/advances/nmy012). This indicates that the milk microbiota could be indirectly influenced by these variations in breast milk composition. Additionally, a higher intake of saturated fatty acids (SFAs) and monounsaturated fatty acids (MUFAs) has been linked to a lower relative abundance of Corynebacterium (DOI: 10.3945/jn.113.176974). Furthermore, Lactobacillus plantarum has been identified as a major component of the breast milk microbiota in diets that include fermented vegetables (DOI: 10.3389/fmicb.2017.00049).

Comments 4. The introduction does not mention factors differentiating the human milk microbiome, i.e.: length of pregnancy, area of â€‹â€‹residence, use of probiotics.

Response 4: Thank you for your valuable feedback. We have modified the manuscript accordingly by adding information about factors that differentiate the human milk microbiome, such as the length of gestation, area of residence, and the use of probiotics, as well as additional relevant factors (lines: 74 - 80).

Comments 5. Why was the analysis of transitional milk omitted?

Response 5: Thank you for your insightful question. The aim of our study was to investigate the core milk microbiota through a cross-species analysis. To achieve this, we focused on analyzing colostrum and mature milk. Colostrum, being the first milk produced postpartum, contains unique bacteria that are not present in mature milk. This initial microbiota plays a crucial role in the early colonization of the infant's gut. By studying colostrum, we can understand the initial bacterial exposure an infant receives. Mature milk, on the other hand, is available for the majority of the lactation period and serves as the primary source of nutrition for the infant. It was essential for our study to compare the microbial composition of mature milk with bovine and goat milk, which are the main alternatives to human milk. Transitional milk, while also important for the neonate, represents an interim phase with a less stable microbial composition that can vary from day to day. Analyzing transitional milk would require thorough monitoring and daily sampling to minimize potential bias. By focusing on the main types of milk, colostrum, and mature milk, we aimed to provide a comprehensive understanding of the core milk microbiome. This approach allowed us to identify key differences and similarities in the microbial communities present in human, bovine, and goat milk.

Comments 6. Lack of precise description of how the milk was obtained. Was the milk collected manually or with a breast pump?

Response 6: The human milk samples were collected using a sterile breast pump. The collection took place in the hospital during the first-morning feeding session, where the newborn was nursed from one breast while a sterile breast pump was used to collect milk from the other breast. Before sample collection, the breast was cleaned, and the nipple was disinfected to avoid contamination. The breast milk samples were collected into sterile tubes. For animals the milk was obtained also using a pump. To clarify the milk collection process, we have revised the methods section of the manuscript (lines 146-166).

Comments 7. Why was milk not collected from animals at the same time points as from women?

Response 7: Thank you for allowing us to clarify this. The milk was not collected from animals at the same time points as from women due to differences in the milking duration for bovines and goats. The lactation period for goats lasts approximately 284 days, while for cows, it is around 305 days. Consequently, a time point was selected for animals and humans corresponding to early mature milk. This timing ensured that the milk samples represented the stages most relevant for infant nutrition and allowed for meaningful comparisons between human and animal milk. This study addresses a less explored aspect, comparing the microbial communities in human breast milk with those in mature milk from species commonly used for milk consumption. Since mature animal milk is widely utilized in infant formulas and for child/adolescent nutrition, we focused on identifying the shared microbial communities between mature animal milk and human colostrum and mature milk. This approach enabled us to better understand the potential microbial contributions of these milk sources to infant and child health. The text has been updated to make this point obvious.  Figure 5 have been updated to make clear the objective of the comparative study performed. That is to identify the core milk microbiome in colostrum and mature breastmilk and its abundance variability compared to the main dairy animals’ mature milk.

Comments 8. The issue of hygiene during the collection of material for testing is debatable. The level of hygiene in women is higher than in animals. Doesn't this affect the results of the study?

Response 8: Thank you for expressing your concern. The animals’ udders were thoroughly sterilized before the sample was collected, thus ensuring that no contamination occurred between the collected sample and the animal’s skin. We have revised the 2.1 section of the manuscript to clarify the milk collection process (lines: 160-166).

Comments 9. The study group is very small, I suggest conducting research on a larger scale. Was the possibility of the influence of factors such as: the woman's intake of probiotics, the length of hospitalization, diseases during pregnancy such as gestational diabetes, taken into account?

Response 9: Thank you for your valuable feedback. We acknowledge that the sample size of our study is small. However, our primary objective was to perform a comparative interspecies study to examine milk's core microbiome. To ensure that a bacterial genus is reliably detected, we set a threshold that required the genera to be present in at least 4 out of 5 samples. Thus, focusing on abundant bacteria present in the majority of samples and applying rigorous filtering during the analysis minimized potential bias in our results. Additionally, we validated our findings by cross-referencing with all available studies in the milk microbiome field using the Microbiome Atlas of Mothers and Infants database, ensuring the robustness and relevance of our results (Table S5). Nonetheless, your suggestion is valid, and a larger sample size could help us identify more statistical differences in abundance analysis, providing a more comprehensive understanding of the human milk microbiome. However, our study retains a significant novelty, and publishing its results at this stage will greatly facilitate our research direction regarding the core milk microbiome of humans.  

Regarding the potential influence of other factors, the intake of probiotics and antibiotics was thoroughly examined, and none of the women had taken either before or during the study period. Milk was collected three days postpartum, which aligns with the mandatory hospital stay policy. None of the women in our study had complications that required an extended hospital stay, thus minimizing any potential impact of hospitalization on the microbial communities of mature milk. All participants were screened to exclude those with gestational diabetes and preeclampsia, ensuring that these conditions did not influence the study results.

Comments 10. In the discussion, please do not repeat the description of your results, which have already been described earlier - in the results section.

Response 10: Thank you for your suggestion. We have revised the discussion section to reflect the interpretation of our results better and make it clearer for the reader. Any unnecessary repetitions have been removed.

Reviewer 2 Report

Comments and Suggestions for Authors

Overall in interesting enough piece of work and nice to see a direct comparison between species. Few suggestions and comments:

1.        I feel the paper is longer than is needed, particularly the results and the discussion section. It goes into a bit of overexplaining, whereas the methodology used is pretty established at this point. Are all the graphs really needed? Do they need to be described that extensively? It gets difficult to point out the key results as they are ‘swimming in an ocean of much else’. I would think the authors need to focus more.

2.        Line 14: milk is not a tissue

3.        Line 18: avoid using ‘that’s’. instead use ‘that is’

4.        Line 59: ‘may be’ or ‘is’?

5.        Line 92: unclear what ‘milk of all 3 species’ refers to

6.        Section 2.1: why were bovine and caprine milk samples taken much further in location and why were not also colostrum and mature milk taken there? Would have been much more insightful.

7.        Line 202: you’re assuming here that cow breed impacts microbiota, but can you support this? References? Other proof?

8.        Figure 1 can be deleted

9.        Paper seems to be missing a conclusions section

Author Response

Comments 1.        I feel the paper is longer than is needed, particularly the results and the discussion section. It goes into a bit of overexplaining, whereas the methodology used is pretty established at this point. Are all the graphs really needed? Do they need to be described that extensively? It gets difficult to point out the key results as they are ‘swimming in an ocean of much else’. I would think the authors need to focus more.

Response 1: Thank you for your constructive feedback. We appreciate your observations regarding the length of the paper, particularly the results and discussion sections. We have revised the results and discussion sections in response to your comments to enhance clarity and focus. We have removed redundant explanations and ensured the key results are highlighted more effectively. Additionally, we have removed Figure 1 and simplified Figure 5 (the old figure 6) to make it easier for researchers to identify the significant outcomes of our research. These changes will improve the readability and impact of the manuscript, and we thank you again for your valuable suggestions.

Comments 2.        Line 14: milk is not a tissue

Response 2: Thank you for pointing out this discrepancy. It has been changed in the manuscript (line 14).

Comments 3.        Line 18: avoid using ‘that’s’. instead use ‘that is’

Response 3: Thank you for noticing this. It has been changed to “that has” (line 19).

Comments 4.        Line 59: ‘may be’ or ‘is’?

Response 4: Thank you for pointing this out. “Is” would be a better choice of words since multiple studies have shown this.

Comments 5.        Line 92: unclear what ‘milk of all 3 species’ refers to

Response 5: Thank you for pointing out this mistake. The structure of the paragraph has been slightly altered to ensure the point is clear (126-140).

Comments 6.        Section 2.1: why were bovine and caprine milk samples taken much further in location and why were not also colostrum and mature milk taken there? Would have been much more insightful.

Response 6: Thank you for the suggestion. The same local dairy farm, Evrofarma, provided both goat and bovine milk. However, all samples in this study, including human milk samples, were collected from the same region in northern Greece, specifically the Evros region. This consistent geographical sourcing helps limit geographical location's influence on the milk microbiome.

While we acknowledge that collecting colostrum and mature milk from the same animals would provide valuable comparisons, our study focused on comparing human milk with mature bovine and goat milk. This focus aligns with our primary objective of understanding the human milk microbiota in relation to other commonly consumed milk sources. Since mature animal milk is frequently used as a supplement in infant formulas and for child/adolescent nutrition, our primary aim was to identify shared microbial communities between mature animal milk and human colostrum and mature milk. This approach enhances our understanding of the potential microbial contributions of these widely used milk sources to infant and child health.

Comments 7.        Line 202: you’re assuming here that cow breed impacts microbiota, but can you support this? References? Other proof?

Response 7: In response to your comment, we have added supporting references to clarify our statement about the impact of breed on milk microbiota. Studies comparing the milk microbiota of different breeds have shown significant differences in general microbial diversity and taxonomy.

One study (DOI: 10.1371/journal.pone.0205054) comparing the milk microbiota of Holstein Friesian (HF) and Rendena (REN) cows demonstrated significant differences According to alpha and Beta-diversity results, the microbial diversity between the two breeds was statistically significant. The study also showed differences in relative abundances and core OTU composition. Another study found differences in the proportions of different genera detected in the milk collected from various sheep breeds, including Merino, Lacaune, and Assaf (DOI: 10.3168/jds.2021-20911). This further supports the notion that breed influences the microbial composition of milk.

Comments 8.        Figure 1 can be deleted

Response 8: Thank you for your suggestion. We agree that Figure 1 is not essential to the manuscript, so we have removed it to help point out the key results more clearly and to make the results section more concise.

Comments 9.        Paper seems to be missing a conclusions section

Response 9: We appreciate your suggestion regarding the conclusions section. We have revised the manuscript to include a conclusions section, summarizing the key findings and their implications (lines 540-552).

Round 2

Reviewer 1 Report

Comments and Suggestions for Authors

Why was the breast disinfected if we are checking the microbiome? One source of the microbiome of breast milk is the mother's skin. Is disinfection right if women do not practice it before each breastfeeding?

Author Response

Comment 1. Why was the breast disinfected if we are checking the microbiome? One source of the microbiome of breast milk is the mother's skin. Is disinfection right if women do not practice it before each breastfeeding?

Response 1: Thank you for your insightful comment. We understand your concern regarding the disinfection of the breast before sample collection, considering the established mechanisms influencing the breast milk microbiome, such as the entero-mammary pathway, the retrograde inoculation pathway, and the resident mammary microbiota. While it is acknowledged that the skin can influence the milk microbiota, it is not considered the primary source of the microbiome in breast milk (10.1016/j.chom.2019.01.011). Commensal bacteria from the skin can colonize the mammary gland; however, it is important to note that the skin microbiota and breast milk microbiota are distinct components of the overall microbiome. Additionally, it is important to recognize that the maternal skin microbiota is crucial for the infant and can influence the infant's skin microbiota (10.1016/j.chom.2023.01.018). For instance, it is supported that the largest contribution to the infant's microbiota was from breast milk (mean 31.6%), followed by maternal skin (25.7%).

The primary reason for disinfection in our study was to minimize the risk of contamination from external, non-resident bacteria, including those from the infant's mouth and surrounding skin, as well as contamination during sampling, that could potentially skew the results and misrepresent the core milk microbiome, as supported by numerus studies (10.3390/nu15030696, 10.1186/s12884-024-06604-x, 10.3390/app11209400, 10.3389/fmicb.2021.557180). This approach was taken to ensure that the bacteria analyzed were indeed representative of those typically present in the milk and not transient contaminants. We revised the manuscript accordingly (lines: 71-72, 79-81, 156-158).

We acknowledge that in a natural breastfeeding scenario, disinfection is not practiced. However, our objective was to obtain a clear and focused understanding of the microbial communities inherently associated with breast milk, minimizing interference from other factors. This controlled approach allows for a more accurate baseline comparison with the microbiomes of mature bovine and goat milk.